# Bridging the Domain Gap: Self-Supervised 3D Scene Understanding with Foundation Models

**Zhimin Chen**[1]
Clemson University
zhiminc@clemson.edu

**Longlong Jing**[2]
The City University of New York
ljing@gradcenter.cuny.edu

**Yingwei Li**[3]
Johns Hopkins University
yingwei.li@jhu.edu

**Bing Li**[✉1]
Clemson University
bli4@clemson.edu

## Abstract

Foundation models have achieved remarkable results in 2D and language tasks like image segmentation, object detection, and visual-language understanding. However, their potential to enrich 3D scene representation learning is largely untapped due to the existence of the domain gap. In this work, we propose an innovative methodology called Bridge3D to address this gap by pre-training 3D models using features, semantic masks, and captions sourced from foundation models. Specifically, our method employs semantic masks from foundation models to guide the masking and reconstruction process for the masked autoencoder, enabling more focused attention on foreground representations. Moreover, we bridge the 3D-text gap at the scene level using image captioning foundation models, thereby facilitating scene-level knowledge distillation. We further extend this bridging effort by introducing an innovative object-level knowledge distillation method that harnesses highly accurate object-level masks and semantic text data from foundation models. Our methodology significantly surpasses the performance of existing state-of-the-art methods in 3D object detection and semantic segmentation tasks. For instance, on the ScanNet dataset, Bridge3D improves the baseline by a notable margin of 6.3%. Code will be available at: https://github.com/Zhimin-C/Bridge3D

## 1 Introduction

In recent years, task-agnostic pre-trained representations have fundamentally reshaped the landscape of Natural Language Processing (NLP), driven by the success of foundation models such as GPT-3 [55], PALM [13], T-NLG [31], and BERT [7]. Parallel advancements have been observed in the realm of computer vision, where foundation models like CLIP [54], Grounding DINO [43], DINOV2 [49], BLIP [39], and SAM [37] have established new benchmarks in 2D vision tasks, emerging as the leading approach for achieving state-of-the-art performance. However, the potential of these powerful models in advancing 3D scene understanding is yet to be fully realized, primarily due to the limited availability of large-scale 3D-text pair datasets and the considerable cost associated with procuring high-quality 3D annotations. Despite recent studies demonstrating the potential of individual foundation models like CLIP [54] or MOCO [27] in enhancing 3D scene understanding, a comprehensive exploration of the utility of other foundation models and their synergistic combinations remains a largely uncharted territory.

To address this challenge, we propose a novel framework **Bridge3D** that harnesses the strengths of multiple foundation models to advance 3D representation learning through a self-supervised learning

37th Conference on Neural Information Processing Systems (NeurIPS 2023).

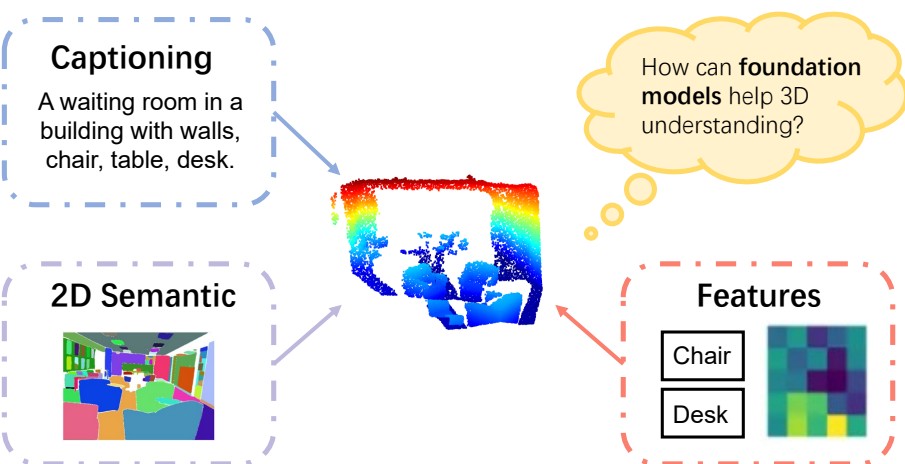

Figure 1: **The motivation of Bridge3D**. The driving force behind Bridge3D is to create a method that bridges the gap between 3D models and text/image foundation models via self-supervised learning.

approach. Specifically, the Bridge3D leverages image captioning outputs from foundation models to generate 3D scene prompts, establishing a bridge between 3D and text domains for scene-level knowledge distillation. These generated prompts are further utilized to produce instance segmentation results with the assistance of Grounding DINO and SAM. Subsequently, we employ a 3D network trained on a self-supervised task that distillates the knowledge from text and 2D to point clouds at the scene level. This is followed by the distillation of these multidimensional features into 3D features at the object level. Additionally, to optimize point reconstruction, we propose an inventive masking and patch-dropping strategy that redirects the model's attention toward foreground object representation learning.

Our proposed framework effectively circumvents three significant hurdles in 3D self-supervised learning. **Firstly**, the adoption of foundation models in our novel masking and dropping strategy allows the network to concentrate more on foreground object representation learning, thereby enhancing the performance of the 3D model. This is a distinct shift from traditional 3D masked autoencoder methods that rely on a random masking strategy and reconstruct all point clouds, which impairs representation learning due to the imbalance between foreground and background points. **Secondly**, the lack of datasets incorporating both 3D and text description pairs significantly hampers the potential for large-language models to contribute to 3D understanding. To overcome this obstacle, Our method first employs image captioning to generate text descriptions from paired images of point clouds, effectively bridging the 3D-text gap at the scene level. This novel integration of 3D and text modalities presents a compelling new frontier for improving self-supervised 3D scene understanding. **Lastly**, our approach stands in contrast to previous methodologies [9; 59] that facilitated either 2D to 3D or text to 3D distillation in isolation due to inherent limitations in mask generation. Instead, our strategy leverages foundation models to generate highly precise object-level masks and semantic text information. This approach seamlessly integrates object-level 3D, visual, and textual features, thereby significantly enhancing the quality of 3D scene representation learning.

We evaluate our method on multiple datasets, including SUN RGB-D [75] and ScanNet [14] for 3D object detection and S3DIS [5] for 3D semantic segmentation. Our approach outperforms state-of-the-art self-supervised learning methods in both tasks, demonstrating the effectiveness of our proposed framework. The contributions of Bridge3D can be summarized as follows:

1. We propose a novel masking and patch-dropping strategy based on foundation models to refine the focus of the network on foreground representation learning for 3D masked autoencoders.

2. We propose a novel scene-level and object-level multi-modality knowledge distillation method that pre-trains a 3D network via features, semantic masks, and captions obtained from foundation models

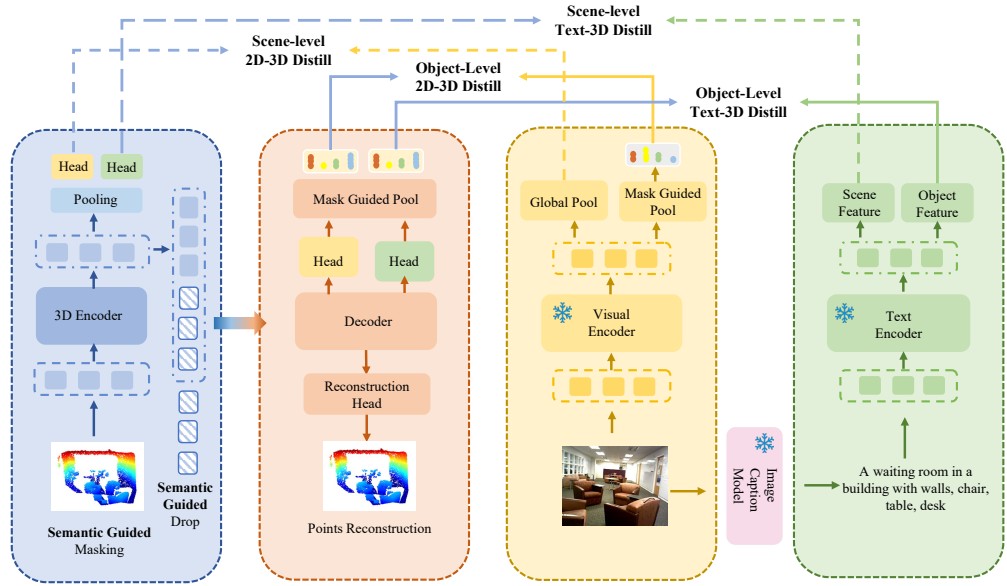

Figure 2: **Overview of Bridge3D**. Our method employs features, semantic masks, and captions derived from foundation models to improve 3D representation learning. We use semantic masks to guide the masking and reconstruction phases in the masked autoencoder, which intensifies the network's attention on foreground objects. At the scene level, we use image captioning foundation models to bridge the scene-level 3D-text gap. Additionally, we facilitate the distillation of well-learned 2D and text representations to the 3D model at the object level by leveraging foundation models to generate accurate object-level masks and semantic text information.

3. To the best of our knowledge, this is the first research to harness multiple foundation models for self-supervised 3D scene understanding. This pioneering approach has been shown to outperform state-of-the-art methods in various downstream tasks.

## 2   Related Work

**3D Self-Supervised Representation Learning.**    Recently, self-supervised pre-training on unlabelled point clouds [2; 24; 12; 66; 73; 35; 22] has shown promising transferable ability, providing a good network initialization for downstream fine-tuning. Several methods have been proposed for pre-training point cloud features, including learning relative position [58], multiple pretext tasks [26], and contrastive learning [17; 33; 57; 65; 72; 3; 35; 23; 21]. Info3D [57] extends the InfoMax and contrastive learning principles to 3D shapes. PointContrast [65] conducts point-level contrast on two transformed views of the same point cloud. Zhang [72] contrasts instance-level representations obtained from the same scenario but processed by different model architectures. CrossPoint [3] introduces an auxiliary multi-modal contrastive objective that captures 3D-2D correspondence, leveraging the complementary attributes of point clouds and images. Point-BERT [67] uses pre-trained tokenizers to indicate discrete point tokens, while Point-MAE [50] applies Masked Autoencoders (MAE) to directly reconstruct the 3D coordinates of masked tokens. Our proposed method uses Point-MAE as the baseline, but leverages foundation models to guide the masking and reconstruction stages. Additionally, we leverage image and text knowledge from foundation models to enhance 3D self-supervised learning.

**Foundation Models.**    The field of AI research has experienced a paradigm shift with the emergence of models trained on massive amounts of data at scale, commonly referred to as foundation models [19; 4; 56; 62; 25; 60]. These models have demonstrated remarkable performance in various language and visual-related tasks. The use of large-scale text pre-training on attention-based models  [15; 71] has led to the increasing popularity of vision-language models (VLM) due to their impressive performance in visual understanding tasks  [54; 46; 40]. Recent advancements in contrastive learning have enabled CLIP  [54] to perform multimodal learning with 400M data crawled from the web. CLIP has been extended for high-efficiency model training and cycle consistency through various methods

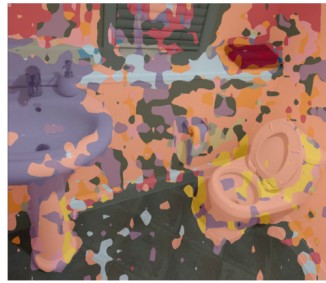 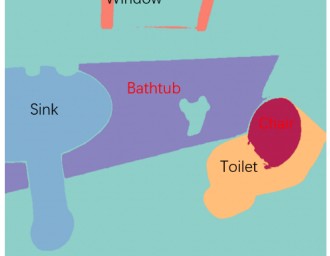 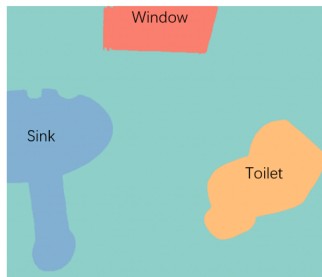

(a) Semantic masks of MaskCLIP with all labels as prompts.

(b) Semantic masks of Ground-SAM with all labels as prompts.

(c) Semantic masks of Ground-SAM with image caption prompts.

Figure 3: **The comparison of zero-shot semantic results.** (a) MaskCLIP failed to perform semantic segmentation accurately, yielding low accuracy overall (b) when all possible labels were used as prompts, the instance segmentation from foundation models results were prone to false positives (as seen in the Chair and Bathtub in this example) (c) by leveraging image captioning model to generates text prompts, the performance of the instance segmentation from Ground-SAM is further improved. This approach is beneficial for 3D scene understanding.

[40; 39; 38]. BLIP [39] includes text-to-image generation as an auxiliary task, which results in better performance by utilizing synthetic data as a bonus. More recently, the success of the foundation models has been achieved in the pure computer vision area. Segment Anything (SAM) [37] has been proposed to act as a generic image segmentation model trained on the large visual corpus. To overcome the drawback of CLIP which overlooks the visual local information, DINOV2 [49] is proposed, which is trained with self-supervised learning and achieves results that match or surpasses the standard approach used in task-specific fields. Grounding DINO [43] extends a closed-set detector DINO to open-set object detection by performing vision-language modality fusion at multiple phases.

**Self-supervised 3D Understanding with Foundation Models.** A number of studies have proposed strategies for knowledge transfer from pre-trained 2D foundation models to 3D representations at the object level [28; 30; 69; 68; 51]. For comprehensive scene understanding, recent efforts have improved 3D point representations by exploiting pixel-point alignments for distillation or contrastive learning [59; 9]. The I2P-MAE [70] approach takes advantage of 2D semantic saliency maps from CLIP [54] to guide masking and facilitate knowledge distillation from 2D to 3D at the instance level. Despite these advancements, most current 3D understanding methodologies employing foundation models focus predominantly on CLIP and distill knowledge through feature-level consistency or contrastive learning [9; 59]. The potential to utilize the capabilities of other foundation models, such as BLIP [39] for image-to-text captioning, SAM [37] for mask generation, Grounding DINO [43] for zero-shot detection, and DINOV2 [49] for high-performance features with detailed localized information, remains largely unexplored. Therefore, we propose to advance self-supervised 3D scene understanding by newly incorporating features, semantic masks, and captions obtained from various foundation models, achieving superior performance compared to state-of-the-art methodologies.

## 3 Methodology

The pipeline of Bridge3D is illustrated in Fig. 2. Our approach employs semantic information and features extracted from well-established foundation models to enhance 3D scene representation learning using self-supervised learning. The proposed method consists of three components: a semantic-guided masked autoencoder, multi-modal scene-level knowledge distillation, and multi-modal object-level knowledge distillation.

### 3.1 Mask Generation by Foundation Models

Our proposed methodology leverages existing foundation models to produce instance segmentation masks. Initially, we use Tag2text [34], based on BLIP [39], to create image captions. We leverage ChatGPT to filter captions objects neither in ScanNet [14] nor SUN RGB-D [61] dataset to generate text prompts for the 3D scene. Subsequently, we employ SAM [37] to generate masks for the image.

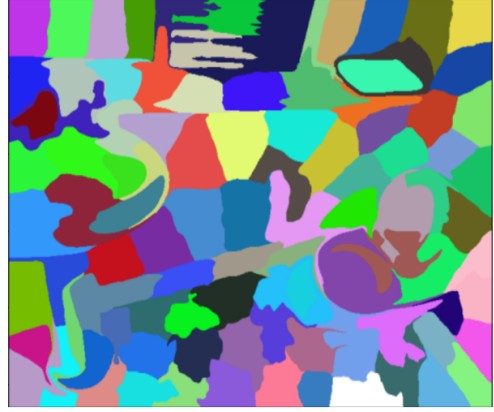 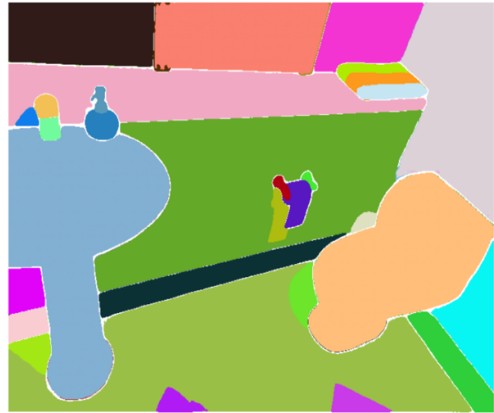

(a) Masking results of SLIC super-pixels.

(b) Masking results by our method which combines Grounding-DINO and SAM.

Figure 4: **The comparison of masks.** (a) Masks generated from super-pixel and (b) Masks generated from our method. Our method can provide much more accurate object-level masks compared to super-pixel and thus benefits the multi-modality knowledge distillation.

Lastly, we use the text labels as prompts for Grounding Dino [43] to create the corresponding bounding box, which is then inputted into SAM to produce both zero-shot instance segmentation labels and segmentation masks $\mathcal{O}_1, \ldots, \mathcal{O}_N$. To establish the dense visual token-point token correspondence $x_i, p_i$, we calibrate the point cloud with the respective images, where $x_i$ and $p_i$ signify the $i$ paired image feature and point feature. This procedure is completed offline and saved locally, with the generated labels being used directly during the self-supervised training stage. As shown in Fig.3 and Fig.4, the instance segmentation masks generated from foundation models outperform previous methods in terms of semantic results and object-level masks. Furthermore, Fig. 3 demonstrates that the performance of the foundation model is further improved when using caption methods as prompts and filtering out 3D-unrelated text.

## 3.2 Semantic Guided 3D Masked Autoencoder

To let the 3D model understand 3D-specific representations, we leverage Point-MAE [50] as the baseline for pre-training, which learns a meaningful representation with the pretext task of recovering the original inputs from visible ones. The Point-MAE [50] method utilizes standard Transformers as the backbone of its architecture, with an encoder-decoder structure that is asymmetric. The encoder takes visible tokens $T^v$ as input and generates encoded tokens $T^e$, while the decoder contains fewer Transformer blocks and is responsible for generating the reconstructed masked 3D coordinates. Positional embeddings are incorporated into each Transformer block to provide location-based information. The encoded tokens $T^e$ are padded with learnable mask tokens $T^m$ and sent to the decoder. A set of positional embeddings is added to each Transformer block in the decoder to provide location information to all tokens. The output of the decoder $H^m$ is then passed through a simple fully connected layer to reconstruct the masked 3D coordinates $P^{pre}$. After that, it restores the coordinates of the points in each masked point patch, and to evaluate the accuracy of the predicted coordinates, it computes the reconstruction loss using $l_2$ Chamfer Distance [18], which is formulated as:

$$\mathcal{L}_{mae} = \frac{1}{M_{mask}} \text{Chamfer}\left(P^{pre}, P^{mask}\right) \tag{1}$$

where $P_{mask}$ represents the ground truth of masked points.

**Foreground-aware Masking and Patch Dropping.** The Point-MAE [50] relies on a random masking strategy, resulting in dispersed attention across the entire image and insufficient focus on foreground objects. This dispersion of attention can lead to the model wasting computational resources on unimportant background elements, thereby leading to weaker learned representations. Moreover, the Point-MAE's approach to reconstructing all masked tokens, including those from the background, may further weaken representation learning as the model overly concentrates on the

background due to the foreground-background imbalance. Although MAE [70] has suggested using 2D semantic saliency maps to guide the masking of point tokens for 3D instances in classification tasks, the problem of effectively guiding 3D scene masking still remains a key research challenge.

To address those problems, we propose a semantic-guided masking strategy based on segmentation results obtained from foundation models. Specifically, for foreground objects obtained from the segmentation mentioned before, we mask a higher percentage $r_f$ of foreground points compared to the whole masking ratio $r_w$. This generates a more challenging reconstruction task and forces the model to focus more on foreground objects. In addition, instead of inputting all patches $\{P_i\}_{i=1}^{M}$ as in Point-MAE, we randomly drop a percentage $r_d$ of background patches to obtain $\{P_i\}_{i=1}^{N}$ . Where $N = (1 - r_d) \times M$. The transformer decoder reconstructs masked point patches by using features from visible tokens and the positional information of both visible patches and mask patches. The patches that are dropped will not be reconstructed by the decoder. With enough background patches dropped, the decoder sees fewer points to perform the trivial up-sampling, which further improves the 3D representation learning and accelerates the pre-training by reducing the input data.

### 3.3 Scene-level Multi-modal Knowledge Distillation

Although some works have investigated scene-level multimodal learning using image and point clouds [36; 11; 41; 70], exploring scene-level multimodal learning with text and point clouds remains a challenge due to the lack of corresponding text descriptions for current 3D scene datasets. To address this problem, we propose to leverage image captioning methods to generate corresponding captions and filter out other 3D irrelevant objects to obtain 3D scene description texts $t_s$. Then, we train the 3D network to align 3D point clouds with their corresponding scene-level images $i_s$ and texts $t_s$. We formulate the proposed method below. Consider a set of $N$ point cloud-image-text pairs $\{F^{3D}, F^{2D}, F^{text}\}$, where $F^{3D}$ represents scene-level point cloud features from the encoder, $F^{2D}$ is the corresponding image features obtained from pre-trained foundation model based on $i_s$, and $F^{text}$ is the text features from pre-trained foundation model based on $t_s$. Our model maps scene-level 3D features $F^{3D}$ to the hidden representation $\hat{F}^m$ for each modality $m$ with a projection head $E_m$. The mapping process can be formulated as:

$$\hat{F}^m = E_m(F^{3D}), \tag{2}$$

Due to the attributes to avoid representation collapsing, previous methods [9; 59] utilize InfoNCE loss [48] to conduct multi-modality knowledge distillation. However, in this work, we find that leveraging positive only $L_1$ smooth loss generates better results. We think this is because those foundation models have learned discriminative features during the pre-training stage, and thus negative pairs are not necessary for the distillation stage. The scene-level distillation between 3D-image features; and 3D-text features are defined by:

$$\mathcal{L}_{scene} = L_1(\hat{F}^{2D}, F^{2D}) + L_1(\hat{F}^{text}, F^{text}) \tag{3}$$

where $L_1$ represents the $L_1$ smooth loss.

### 3.4 Object-level Multi-modal Knowledge Distillation

While the value of object-wise feature representations in downstream tasks like semantic segmentation and detection is well-proved [74; 29; 64; 6], the generation of unsupervised masks presents a substantial challenge. Traditional techniques in computer vision, such as Felzenszwalb-Huttenlocher [20] and super-pixel [1], have been employed in earlier methods [59]. However, these techniques yield subpar masking results and cannot generate semantic labels, which hinders their ability to bridge the 3D-2D-text gap and leaves the potential of powerful language foundation models unrealized. The recent CLIP2Scene method [9] uses the MaskClip [76] model to generate dense semantic predictions, but it falls short of generating instance semantic results, which prevents object-level visual representations from being distilled into 3D models. The inferior quality outputs of MaskClip, Felzenszwalb-Huttenlocher, and super-pixel impede 3D representation learning. In contrast, our proposed Bridge3D method leverages high-quality masks obtained from foundation models to guide object-level knowledge distillation, thereby enhancing 3D scene understanding.

In the object-level knowledge distillation phase of Bridge3D, we propose a distinctive approach to multi-modality distillation through reconstructed tokens from the decoder, as opposed to previous

methods [9; 59] that directly distill the knowledge from other modalities to 3D post-encoder. The Point-MAE uses the decoder to reconstruct these masked tokens and learn specific features. In our method, we not only task the decoder with reconstructing masked point clouds but also reconstruct text and visual features that correlate with visible point tokens to distill the multi-modal knowledge. We do not reconstruct text and visual features of masked tokens that are dropped in the point reconstruction part, as mentioned in Sec. 3.2. To be specific, $I_i$ represents the visual features obtained from the pre-trained visual model and belonging to the mask $O_i$. We use the mapping function to group the points into corresponding masks: $\mathcal{G}_1, \ldots, \mathcal{G}_k$. For each scene, we compute mask visual features and mask point features by average pooling:

$$f_{l,i}^{2D} = \frac{1}{\mathcal{O}_i} \sum_{i \in \mathcal{O}_j} (I_j) \tag{4}$$

$$\hat{f}_{l,i}^{2D} = \frac{1}{\mathcal{G}_i} \sum_{j \in \mathcal{G}_i} (\mathcal{F}_{2D}(H_i)) \tag{5}$$

Where $H_i$ represents visible tokens from the decoder, $\mathcal{F}_{2D}$ is the projection head. For the text-to-point cloud knowledge distillation, as only foreground objects have the text information from the semantic labels, we choose corresponding foreground masks $\mathcal{J}_1, \ldots, \mathcal{J}_s$ from $\mathcal{G}$. Text features are obtained following:

$$f_{l,i}^{text} = \phi_{text}(t_{s,i}) \tag{6}$$

$$\hat{f}_{l,i}^{text} = \frac{1}{\mathcal{J}} \sum_{j \in \mathcal{J}_i} (\mathcal{F}_{text}(H_j)) \tag{7}$$

Where $t_{s,i}$ is the corresponding mask semantic label and $\phi_{text}$ is the pre-trained text encoder, and $\mathcal{F}_{text}$ is the projection head. We then transfer visual-text pairs to point-text pairs $(f_{l,i}^{3D}, f_{l,i}^{text})$ and distill the knowledge from text to the point cloud in the object-level. The objective function is as follows:

$$\mathcal{L}_{object} = \frac{1}{K} \sum_{i}^{K} L_1(\hat{f}_{l,i}^{2D}, f_{l,i}^{2D}) + \frac{1}{S} \sum_{i}^{S} L_1(\hat{f}_{l,i}^{text}, f_{l,i}^{text}) \tag{8}$$

The $L_1$ is the smooth $L_1$ loss. Our final loss is the sum of previous loss terms.

$$L_{final} = L_{mae} + L_{scene} + L_{object} \tag{9}$$

## 4 Experiments

In this section, we first introduce the pre-training setting of Bridge3D. Then, we show the effectiveness of our method on several popular downstream tasks, including 3D object detection and 3D semantic segmentation. Finally, we conduct extensive ablation studies to show the effectiveness of each design. We put more details into the *supplementary materials*.

### 4.1 Self-supervised Pre-training

**Network architectures.** For the 3D backbone encoder, we utilize the same architecture as Point-MAE [50]. For the image branch, we follow DINOV2 ViT-B [49] to divide 518x518 images into regular patches with a size of $37 \times 37$, before the ViT backbone. For the image branch, we directly utilize the CLIP ViT-B [54] to extract text features. For image captioning, we leverage Tag2text [34].

**Pre-training.** During this stage, we perform training of the model for 120 epochs by employing the ScanNet dataset [14] consisting of point clouds and their corresponding images. For the text prompts, we only utilize all class names of ScanNet and SUN RGB-D as the prompts and filter other classes generated from image captioning. We use AdamW [45] optimizer with a base learning rate of 5e-4 and weight decay of 5e-2, along with a batch size of 64. The whole masking ratio $r_w$ is set to 70% and the drop ratio $r_d$ is set to 40%. The cosine learning rate scheduler is applied, with a drop path rate and warm-up epochs set to 0.1 and 10, respectively. The encoder depth is set to 6, and we utilize the same decoder as Point-MAE [50], with the decoder depth set to 2.

| Methods | Pre-trained | SUN RGB-D | | ScanNetV2 | |
| --- | --- | --- | --- | --- | --- |
| | | $AP_{25}$ | $AP_{50}$ | $AP_{25}$ | $AP_{50}$ |
| VoteNet [52] | *None* | 57.7 | 32.9 | 58.6 | 33.5 |
| PointContrast [65] | ✓ | 57.5 | 34.5 | 59.2 | 38.0 |
| Hou et al. [32] | ✓ | - | 36.4 | - | 39.3 |
| 4DContrast [10] | ✓ | - | 38.2 | - | 40.0 |
| DepthContrast [72] | ✓ | 61.6 | 35.5 | 64.0 | 42.9 |
| DPCo [41] | ✓ | 60.2 | 35.5 | 64.2 | 41.5 |
| 3DETR [47] | *None* | 58.0 | 30.3 | 62.1 | 37.9 |
| +Bridge3D(from scratch) | *None* | 57.6 | 31.9 | 61.1 | 38.6 |
| +Point-BERT[67] | - | - | - | 61.0 | 38.3 |
| +Point-MAE [50] | ✓ | - | - | 63.4 | 40.6 |
| +MaskPoint [42] | ✓ | - | - | 63.4 | 40.6 |
| +ACT [16] | ✓ | - | - | 63.5 | 41.0 |
| +PiMAE [8] | ✓ | 59.9 | 33.7 | 63.0 | 40.2 |
| +Bridge3D | ✓ | **61.8(+3.8)** | **35.9(+5.6)** | **65.3(+3.2)** | **44.2(+6.3)** |
| GroupFree3D [44] | *None* | 63.0 | 45.2 | 67.3 | 48.9 |
| +Bridge3D(from scratch) | *None* | 62.2 | 45.0 | 66.1 | 48.3 |
| +Point-MAE [50] | ✓ | 63.9 | 46.1 | 67.4 | 49.8 |
| +PiMAE [8] | ✓ | 65.0 | 46.8 | 67.9 | 50.5 |
| +Bridge3D | ✓ | **67.9(+4.9)** | **48.5(+3.3)** | **69.1(+1.8)** | **51.9(+3.0)** |

Table 1: **3D object detection results on ScanNet and SUN RGB-D dataset.** We adopt the average precision with 3D IoU thresholds of 0.25 ($AP_{25}$) and 0.5 ($AP_{50}$) for the evaluation metrics.

| Methods | Pre-trained | S3DIS | | ScanNetV2 | |
| --- | --- | --- | --- | --- | --- |
| | | $mIoU$ | $mAcc$ | $mIoU$ | $mAcc$ |
| SR-UNet [65] | *None* | 68.2 | 75.5 | 72.1 | 80.7 |
| PointContrast [65] | ✓ | 70.9 | 77.0 | 74.1 | 81.6 |
| DepthContrast [72] | ✓ | 70.6 | - | 73.1 | - |
| Hou et al. [32] | ✓ | 72.2 | - | 73.8 | - |
| Standard Transformer [67] | *None* | 60.0 | 68.6 | - | - |
| PointBert [67] | ✓ | 60.8 | 69.9 | - | - |
| PViT [53] | *None* | 64.4 | 69.9 | - | - |
| PViT+Pix4Point [53] | ✓ | 69.6 | 75.2 | - | - |
| Ours(from scratch) | *None* | 61.1 | 67.2 | 67.3 | 73.1 |
| +Point-MAE [50] | ✓ | 64.8 | 70.2 | - | - |
| +Bridge3D | ✓ | **70.2 (+9.1)** | **76.1(+8.9)** | **73.9(+6.6)** | **80.2(+7.1)** |

Table 2: **3D semantic segmentation results on S3DIS dataset.** We adopt the mean accuracy (mAcc) and mean IoU (mIoU) for the evaluation metrics.

## 4.2 Results on Downstream Tasks

For fine-tuning downstream tasks, we discard decoders in pre-training and append task-specific decoders onto the encoder for different tasks.

**Object Detection.** To demonstrate the generality of the proposed method, we also pre-train it on the indoor ScanNetV2 dataset [14] and subsequently fine-tune our method on the object detection task in ScanNetV2 dataset and SUN RGBD [75]. We report our performance on indoor 3D detection based on SOTA methods 3DETR [47] and GroupFree3D [44]. The Table 1 indicates that Our method achieves 66.3 $AP_{25}$ (+4.2) and 45.5 $AP_{50}$ (+7.6) compared to the baseline 3DETR on the ScanNetV2 dataset and also brings significant improvements to both models, surpassing previous baselines consistently in all other datasets and criteria. These experiments' results showcase our method's

| Point-MAE | Semantic Guided Reconstruction | Scene-level Distillation | Object-level Distillation | ScanNetV2 | | S3DIS | |
|---|---|---|---|---|---|---|---|
| | | | | $AP_{25}$ | $AP_{50}$ | $mIoU$ | $mAcc$ |
| ✓ | | | | 62.3 | 39.9 | 64.8 | 70.2 |
| ✓ | ✓ | | | 63.2 | 41.1 | 66.2 | 71.1 |
| ✓ | ✓ | ✓ | | 64.4 | 43.0 | 68.4 | 73.7 |
| ✓ | ✓ | ✓ | ✓ | **65.3** | **44.2** | **70.2** | **76.1** |

Table 3: **The effectiveness of each component.** Ablation study on the effectiveness of each component on 3D object detection and semantic segmentation tasks.

| Text | Image | ScanNetV2 | | S3DIS | |
|---|---|---|---|---|---|
| | | $AP_{25}$ | $AP_{50}$ | $mIoU$ | $mAcc$ |
| | | 62.1 | 37.9 | 61.1 | 67.2 |
| ✓ | | 64.2 | 42.5 | 67.8 | 74.1 |
| | ✓ | 64.7 | 43.3 | 68.3 | 74.5 |
| ✓ | ✓ | **65.3** | **44.2** | **70.2** | **76.1** |

Table 4: **The effectiveness of each modality.** Ablation study on the effectiveness of each modality on 3D object detection and semantic segmentation tasks.

effectiveness in learning superior 3D representations for object detection, highlighting its potential to benefit a wide range of 3D applications.

**Semantic Segmentation.** In Tab. 2, we present the semantic segmentation results on the S3DIS dataset. Despite their efforts, prior 3D self-supervised methods such as PointContrast[65] and [72] only achieved marginal improvements post-pre-training (+2.7 and +2.4 in $mIoU$). Conversely, Pix4Point [53], utilizing a pre-trained 2D model, demonstrated significant progress compared to training from scratch. Most notably, our proposed method incorporates multiple foundation models during pre-training, elevating the metrics by 10.0 and 10.3 respectively, markedly surpassing other state-of-the-art 3D self-supervised methods. These results substantiate the effectiveness of utilizing multiple foundation models for enhancing 3D representation learning for semantic segmentation.

### 4.3 Ablation Studies

**The effectiveness of Each Component.** As shown in Table 3, the results indicate that each component, including the foreground-aware masking strategy, multi-modal scene-level knowledge distillation, and multi-modal object-level knowledge distillation, contributes to better results. More-over, when we combined all components, our proposed method achieved the best performance. The foreground-aware masking strategy proved to be important as it enhanced the learning of foreground object representations in the 3D masked autoencoder. The multi-modal scene-level knowledge distillation, which leverages an image captioning model to generate text descriptions from paired images of point clouds, helped bridge the gap between 3D and text at the scene level. The multi-modal object-level knowledge distillation, which uses foundation models to generate accurate object-level masks and semantic text information, bridged the gap between 3D, 2D, and text object-level features. Overall, our ablation study demonstrates the effectiveness of each component in our proposed framework and highlights the importance of leveraging foundation models to improve 3D scene representation learning.

**The effectiveness of Each Modality.** Table 4 provides a clear insight into the significant contribution of each modality to the overall performance of our method. By integrating all modalities, our method realizes its full potential, showcasing the best results. However, it's worth noting the inherent resilience our method exhibits to modality variations. This adaptability implies that even when the modality mix is altered or reduced, the system remains relatively unaffected in terms of its output quality. This inherent resilience not only underscores the robust architecture of our model but also offers users the freedom to customize the framework based on their specific requirements. The adaptable nature of modality inclusion thus ensures that our method remains both versatile and efficient, enabling users to balance computational overhead with optimal performance.

| Semantic Mask Method | ScanNetV2 | | S3DIS | |
| --- | --- | --- | --- | --- |
| | $AP_{25}$ | $AP_{50}$ | $mIoU$ | $mAcc$ |
| Point-MAE [50] | 62.3 | 39.9 | 64.8 | 70.2 |
| MaskCLIP [76] + Superpixels [1] | 64.1 | 43.0 | 67.1 | 73.4 |
| Bridge3D | **65.3** | **44.2** | **70.2** | **76.1** |

Table 5: **The effectiveness of mask.** Ablation study on unsupervised mask methods on 3D object detection and semantic segmentation tasks.

| Methods | Pre-trained | SUN RGB-D | | ScanNetV2 | |
| --- | --- | --- | --- | --- | --- |
| | | $AP_{25}$ | $AP_{50}$ | $AP_{25}$ | $AP_{50}$ |
| CAGroup3D [63] | *None* | 66.8 | 50.2 | 75.1 | 61.3 |
| +Bridge3D (scene & object level distillation) | ✓ | **68.7** | **52.1** | **76.3** | **62.2** |

Table 6: **The performance on SOTA method.** 3D object detection results on ScanNet and SUN RGB-D dataset based on CAGroup3D.

**The Effectiveness of Masks from Foundation Models.** In Table 5, we conduct a comparative study of mask generation strategies employed by our method and those utilized by prior works. For instance, CLIP2Scene[9] employs MaskCLIP [76] to produce semantic masks. However, this approach fails to provide instance segmentation masks and is solely capable of guiding text-to-3D knowledge distillation. Alternatively, SLidR [59] leverages superpixels [1] for generating object-level masks. Despite this, the superpixels lack semantic information, limiting their use to guiding 2D-to-3D knowledge distillation. In contrast, our method generates instance semantic information, enhancing the functionality and accuracy of the masks. For a comparative analysis, we combine superpixels and MaskCLIP to direct both 2D-to-3D and text-to-3D distillation, essentially mimicking the fusion of CLIP2scene and SLidR. The experimental results reveal that masks generated through our method, which leverages foundation models, yield substantial improvements compared to the combined use of MaskCLIP and superpixels.

**Apply Bridge3D in SOTA Method.** Our method's adaptability extends to its successful application to the recent state-of-the-art (SOTA) work, CAGroup3D [63]. While the network structure made a direct application of the plain transformer challenging, we devised a unique adaptation, pre-training the CAGroup3D's backbone with our scene and object-level distillation, excluding the reconstruction part. Table 2 illustrates how our pre-training approach can enhance and benefit current SOTA 3D detection methodologies, showcasing the potential reach of our technique. Importantly, it should be noted that our framework is optimized for plain transformer-based backbones, and thus the application of Bridge3D to alternative backbones may lead to a reduction in performance.

## 5   Conclusion

In this work, we introduce a pioneering method Bridge3D that capitalizes on foundation models to overcome the hurdles in self-supervised 3D learning. This innovative approach not only enables more focused learning on foreground object representation but crucially bridges the domain gap between 3D and text at the scene level. Furthermore, it enriches the quality of 3D scene representation learning by generating highly accurate object-level masks and semantic textual information, effectively bridging the gap between 3D, 2D, and text object-level features. Our comprehensive experimental results corroborate the superior effectiveness of our approach in amplifying 3D scene understanding. However, the current work primarily focuses on indoor 3D scene understanding, which constitutes a limitation. Looking ahead, we plan to broaden the applicability of Bridge3D to encompass a more diverse set of 3D tasks, including outdoor scene understanding and open-vocabulary 3D tasks. Our work is expected to have no negative societal implications.

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
