# Supplementary Material for Bridging the Domain Gap: Self-Supervised 3D Scene Understanding with Foundation Models

## 1 Baseline: Point-MAE

**Point Patches Generation and Masking:** Following Point-BERT [12], the Point-MAE divides the input point cloud into $n$ irregular point patches (may overlap) via Farthest Point Sampling (FPS) and K-Nearest Neighborhood (KNN) algorithm. The masking strategy is set to random and the mask ratio $m$ is 60 %.

**Embedding:** To embed each masked point patch, the Point-MAE method substitutes it with a mask token that is learnable and weighted-shared. Meanwhile, for unmasked point patches (i.e., those that are visible), Point-MAE employs a lightweight PointNet [8] to extract features from the point patches. The visible point patches $P^v$ are hence embedded into visible tokens $T^v$:

$$T^v = \text{PointNet}\left(P_v\right) \tag{1}$$

**Backbone:** The backbone of Point-MAE is entirely based on standard Transformers, with an asymmetric encoder-decoder. The encoder takes visible tokens $T^v$ as input to generate encoded tokens $T^e$. In addition, Point-MAE incorporates positional embeddings into each Transformer block, thereby adding location-based information. The decoder is similar to the encoder but contains fewer Transformer blocks. The Point-MAE pads encoded tokens $T^e$ with learnable mask tokens $T^m$ and sends them to the decoder. A complete set of positional embeddings is added to every Transformer block in the decoder part to provide location information to all the tokens. The outputs of the decoder are fed to a simple fully connected (FC) layer to reconstruct the masked 3D coordinates. The encoder-decoder structure is formulated as:

$$T^e = Encoder\left(T^v\right) \tag{2}$$

$$H^m = Decoder\left(\text{concat}\left(T^e, T^m\right)\right) \tag{3}$$

The projection head is formulated as:

$$P^{pre} = \text{Reshape}\left(FC\left(H^m\right)\right) \tag{4}$$

**Reconstruction Target:** Point-MAE's reconstruction task aims to restore the coordinates of the points in each masked point patch. To evaluate the accuracy of the predicted coordinates of the masked patches, Point-MAE computes the reconstruction loss by $l_2$ Chamfer Distance [4], which is formulated as:

$$\mathcal{L}_{MAE} = \frac{1}{M_{mask}} \text{Chamfer}\left(P^{pre}, P^{mask}\right) \tag{5}$$

where $P_{mask}$ represents the ground truth of masked points.

Submitted to 37th Conference on Neural Information Processing Systems (NeurIPS 2023). Do not distribute.

| Distillation Metric | ScanNetV2 | | S3DIS | |
|---|---|---|---|---|
| | $AP_{25}$ | $AP_{50}$ | $mIoU$ | $mAcc$ |
| InfoNCE | 65.8 | 45.0 | 70.7 | 76.8 |
| Cosine Similarity | 65.6 | 44.4 | 70.3 | 76.3 |
| $\mathcal{L}_2$ Distance | 65.2 | 44.9 | 69.7 | 75.9 |
| Smooth $\mathcal{L}_1$ | **66.3** | **45.5** | **71.1** | **77.5** |

Table 1: Ablation study on the Distillation metric for 3D object detection and semantic segmentation tasks.

## 2   Implementation Details of Methodology

**Object Detection on ScanNet.**   For the 3D object detection task, We fine-tune our method on the ScanNetV2 [3] dataset based on GroupFree3D [5] and 3DETR [6]. This dataset includes 1,513 indoor scenes with 18 categories of axis-aligned 3D bounding boxes, where 1,201 are for training and 312 are for validation. We utilized the same encoder architecture in the pre-trained stage and the same decoder as in 3DETR and GroupFree3D. For the encoder, we randomly sample 40K points and divided them into 512 patches with 128 points. We train our method the 3DETR for 1,080 epochs with a learning rate of 1e-5. We train our method the GroupFree3D for 1,080 epochs with a learning rate of 6e-5 and a batch size of 8.

**Object Detection on SUN RGB-D.**   We fine-tune our method on the SUN RGB-D [11] dataset based on GroupFree3D [5] and 3DETR [6]. SUN RGB-D contains more than 10, 000 indoor scenes while 5285 for training and 5050 for validation. We utilized the same encoder architecture in the pre-trained stage and the same decoder as in 3DETR and GroupFree3D. For the encoder, we randomly sample 40K points and divided them into 512 patches with 128 points. We train our method the 3DETR for 1,080 epochs with a learning rate of 1e-5. We train our method the GroupFree3D for 1,080 epochs with a learning rate of 3e-5 and a batch size of 8.

**Semantic Segmentation on S3DIS.**   For the 3D semantic segmentation task on the S3DIS dataset [1], we followed standard practice and reserved area 5 for testing while using the remaining areas for training. We utilized a two-layer MLP to project patch features to 96 channels for generating point-wise semantic predictions in the decoder. The patch features were up-sampled using nearest neighbor up-sampling, and the five nearest key points for each target coordinate were concatenated with their distance to the target coordinate. The concatenated features were then projected to 96 channels using a two-layer MLP, and features were aggregated using a weighted sum based on their inverse distance to the target coordinate. Finally, an MLP with a dropout rate of 0.5 was used for classification. To adhere to previous work [9], we voxel downsampled the point clouds with a voxel size of 0.04m and applied the same data augmentation method. For the encoder, we randomly sampled 24K points and divided them into 512 patches with 64 points. We fine-tuned our method for 300 epochs with a learning rate of 1e-3 and a batch size of 8.

## 3   Additional Ablation Studies

**Ablation Study on the Distillation Metric.**   In the ablation study on the contrastive metric, Table 1 shows that our method achieves the best results with the Smooth $l_1$ loss, unlike previous methods [2; 10] that utilize InfoNCE[7] for contrastive learning with positive and negative samples. We argue that this is because our method uses a masked autoencoder in the pre-training stage, which masks a large portion of input tokens, leading to small matched pairs for contrastive learning and decreased performance of InfoNCE loss. Furthermore, the foundation models (DINOV2, CLIP) used in our method are trained with contrastive learning and have already learned discriminative representations, making InfoNCE unnecessary for increasing the distance between positive and negative samples in the distillation stage.

**Ablation Study on the Masking Ratio.**   In our comprehensive ablation study, we analyzed the influence of various masking ratios on the performance of the Bridge3D model in 3D object detection and semantic segmentation tasks. The results depicted in Table 2 disclose that optimal latent feature

| Mask Ratio | ScanNetV2 | | S3DIS | |
|---|---|---|---|---|
| | $AP_{25}$ | $AP_{50}$ | $mIoU$ | $mAcc$ |
| 80% | 65.9 | 44.8 | 70.6 | 76.2 |
| 70% | **66.3** | **45.5** | **71.1** | **77.5** |
| 60% | 65.8 | 45.1 | 71.0 | 77.2 |
| 50% | 65.1 | 44.5 | 70.5 | 76.7 |

Table 2: Ablation study on masking ratio for 3D object detection and semantic segmentation tasks.

extraction is achieved when the masking ratio is set at 70%. Importantly, our experiment also exhibits the robustness of our proposed methodology, maintaining consistent performance across a range of masking ratios. This consistency underscores the wide applicability and efficacy of the Bridge3D framework in learning robust representations of 3D point clouds.