# OpenReview forum: "Bridging the Domain Gap: Self-Supervised 3D Scene Understanding with Foundation Models"
_NeurIPS.cc/2023/Conference — NeurIPS 2023 poster_

### Official Review · Reviewer_MtnQ · 2023-07-06

**Soundness:** 2 fair
**Presentation:** 2 fair
**Contribution:** 1 poor
**Rating:** 5
**Confidence:** 4

**Summary:**

The paper proposes a pre-training strategy to distill the knowledge for large image foundation models. The method proposed is based on a mask auto-encoder transformer model and the setup contains several modules or losses. First, the masking strategy is guided by segmentation labels obtained using foundation models. Moreover, two additional losses at the scene and object level are incorporated to distill information from image foundation models. Lastly, a loss is used to align features from the point encoder to a text encoder.

**Strengths:**

The paper proposes a combination of several foundation models with different modalities, such as segmentation masks, unsupervised pre-trained features, and text embeddings. Moreover, the paper is easy to follow.

**Weaknesses:**

- The novelty of the methods is questionable since other methods have used text foundation models (Language-Grounded Indoor 3D Semantic Segmentation in the Wild), and image pre-trained models (OpenScene: 3D Scene Understanding with Open Vocabularies), or both (ULIP: Learning a Unified Representation of Language, Images, and Point Clouds for 3D Understanding). The paper proposes to combine similar ideas, but they are not new and not enough comparisons to those other pre-training strategies are provided.

- To use an informed mask guidance has been proposed before and the paper misses to mention all related research in this area. Moreover, the proposed masking method is not particular to 3D data and these other strategies for images could be used instead. These papers should be cited and compared to these approaches.
	SemMAE: Semantic-Guided Masking for Learning Masked Autoencoders
	What to Hide from Your Students: Attention-Guided Masked Image Modeling

- The evaluation in general is limited and the only conclusion the reader can draw from these results is that Bridge3D is a better pre-training strategy than vanilla Point-MAE (which is not surprising), and PiMAE. The other comparisons provided in the table are comparing apples and oranges. Most of the methods in Table 1 use VoteNet as an object detection framework, a much older framework than the ones used with Bridge3D. Is the improvement coming from the pre-training or from the newer detection frameworks? Moreover, some of the numbers in the table are not correct. For example, DepthContrast provides results for two object detection frameworks and architectures, VoteNet and H3DNet. However, only the results for VoteNet are presented which are much lower than H3DNet (64.0/42.9/61.6/35.5 vs 69.0/50.0/63.5/43.4). Moreover, these methods use different architectures which makes the pre-training strategies not comparable. In order to claim superiority over all these pre-training strategies the same model and framework for object detection should be used. Since, the proposed method is based on MAE and therefore limits the architecture choice, at least the performance of the proposed method with a VoteNet framework (and other methods with the newer object detection frameworks) should be reported.

Regarding semantic segmentation, the evaluation has also limitations. First, only results for S3DIS are provided. This data set is small and relatively large models overfit very easily, most of the recent improvements in the data set are obtained by the type of data augmentations used. I would suggest the authors to report results on a large data set such as ScanNet, which, although not perfect, is more representative of this task. Moreover, there are other works missing in Table 2 that appear in Table 1, for example [27]. They report 72.2 mIoU which is higher than the one obtained with the proposed method, Bridge3D.

**Questions:**

I do not have any questions for the authors. I believe the paper needs to significantly increase the evaluation to show which is the real contribution wrt previous work and to support the claims of the paper.

**Limitations:**

The paper does not analyze properly the limitations of the method. For example, authors could discuss: some of the foundation models used are trained with annotated segmentation masks. Does this pose a problem for the test sets used on the paper? how similar are the images used for training these foundation models to the images used during pre-training and downstream tasks?

---

> ### Author Rebuttal · Authors · 2023-08-09
>
> With gratitude for the reviewer's valuable insights, we will make appropriate revisions to our paper. In the following section, we address each of the reviewer's questions and comments in detail.
>
> **W1: The novelty of this method**
>
> **1, Difference to the Language-Grounded Indoor 3D Semantic Segmentation in the Wild and OpenScene**. Unlike those previous works that focus solely on distilling either text to 3D or images to 3D, our method innovatively integrates both 2D visual information and textual descriptions to enrich 3D representations. In this paper, we have compared with a very similar and recent CVPR 2023 paper, CLIP2Scene [9] in relative work and ablation study sections. CLIP2Scene paper explores the transfer of CLIP knowledge from 2D image-text pre-trained models to a 3D point cloud network, which aligns closely with our focus. We will also discuss those two papers in relative works in the final version.
>
> **2, Differnce to the ULIP**.
> The ULIP method is specifically tailored for pre-training on instance-level datasets such as ModelNet40 and ScanObjectNN and is hard to be directly applied to 3D scene understanding tasks. Furthermore, our method overcomes the limitations of ULIP that need text labels to generate text features. We leverage powerful image captioning models to generate scene text descriptions and utilize CHATGPT to refine these texts without requiring annotations.
>
> **3, Novelty**.
> To the best of our knowledge, our work first harnesses multiple foundation models for self-supervised 3D scene understanding. The novelty of our approach can be summarized into three key areas: (1) We introduce a pioneering masking and patch-dropping strategy founded on foundation models. Unlike previous 3D masking methods that were confined to instance-level 3D objects, our approach breaks new ground by extending its applicability to 3D scene understanding. (2) We uniquely explore the utilization of highly accurate masks, extracted from state-of-the-art foundation models like SAM and Grounding Dino, a direction not previously pursued in existing works. (3) Our research crafts a novel scene-level and object-level multi-modality knowledge distillation method. This method is innovative in its approach to pre-training a 3D network, drawing upon features, semantic masks, and captions derived from foundation models, forging a new path in the field of 3D representation learning.
>
> **W2: Unreferenced informed mask guidance methods**
>
> Thank you for highlighting previous work in informed mask guidance such as SemMAE and What to Hide from Your Students. We will cite those two papers in the revised version. In our work, we have compared our work with a
> CVPR 2023 paper I2P-MAE [63] in relative works. The I2P-MAE employs 2D semantic saliency maps from CLIP to guide 3D masking strategies. This approach shares commonalities with SemMAE and "What to Hide from Your Students" but diverges when applied to our context. SemMAE, What to Hide from Your Students, and I2P-MAE leverage self-attention of class tokens to obtain masks in object-centric data. However, our research is designed for scenes that contain multiple objects, making those self-attention-based methods inapplicable. This differentiation indicates the novelty of our approach.
>
> **W3: The limitation of evaluation**
>
> Due to the character limitation, we put this part in the global response.
>
> **W4: 3D segmentation results in ScanNet**
>
> Recent plain transformer-based works such as Pixel4Point[47] and ACT[1] only evaluate their results in the S3DIS dataset. However, we acknowledge your concerns regarding the S3DIS dataset's limitations. To evaluate the versatility of our approach, we evaluate our method in  ScanNet segmentation tasks.  Table 1 demonstrates that Bridge3D achieves SOTA performance in segmentation tasks on ScanNet even starting from a comparatively lower baseline. It should be notice that SR-UNet is the backbone designed specifically for segmentation, but the plain transformer is a more general backbone that can be applied in both detection and segmentation.
>
> **Table 1: 3D semantic segmentation results on ScanNet dataset.**
>
> | Methods          | Pre-trained  | $mIoU$                  |
> |------------------|--------------|-------------------------|
> | SR-UNet [58]        | *None*       | 72.2                    |
> | DepthContrast [65]    | ✓            | 73.1                    |
> | Hou et al. [27]      | ✓            | 73.8                    |
> | PointContrast [58]   | ✓            | **74.1(+1.9)**          |
> | Ours(from scratch)| *None*       | 67.3                    |
> | +Bridge3D        | ✓            | **73.9(+6.6)**          |
>
> **W5: Missing works**
>
> Thank you for bringing our attention to the discrepancy between Tables 1 and 2, specifically with regard to work [27]. The absence of [27] was an unintentional omission, and we regret any confusion this may have caused. It is important to note that the mIoU of 72.2 reported by [27] is utilizing a more powerful baseline. The improvement we achieve with our method over the transformer baseline is substantially greater compared to other pre-training methods.
>
> ****W6: Limitations of the paper are not discussed****
>
> We also put this part to the global response part.
>
> As for the question about foundation models, all foundation models used in our method do not leverage data from downstream tasks for training. This ensures an unbiased approach to downstream tasks. Your inquiry into the data similarities among foundation models and downstream tasks is a thoughtful and pertinent one. However, it is hard to evaluate their similarities due to the large scale of data used for foundation model training.
>
> [1] Dong, Runpei, et al. "Autoencoders as Cross-Modal Teachers: Can Pretrained 2D Image Transformers Help 3D Representation Learning?." The Eleventh International Conference on Learning Representations. 2022.

---

> > ### Comment · Reviewer_MtnQ · 2023-08-17
> >
> > First, I would like to thank the authors for the thorough rebuttal and for providing numbers for numerous experiments. From these new experiments, readers are able to assess the improvement introduced by the method. Regarding the mask guidance, thanks for including this in the final version of the paper. I agree with the authors that these methods cannot be easily transferred to multiple object images. Regarding novelty, my concerns still remain. Combining several foundation models does not seem a big contribution when those have been used separately before. Also, ULIP uses images and text for pre-training, even if it was at object level. The proposed method uses the same ideas at object level. However, since many of my concerns regarding evaluation have been addressed I will increase my score.

---

> > > ### Author Response · Authors · 2023-08-17
> > > **Thanks for the revised score and addressing concerns regarding novelty**
> > >
> > > Thank you for your thoughtful review and for recognizing the efforts we've made in addressing your previous concerns about evaluation. We're pleased that the additional experiments and the inclusion of mask guidance have helped clarify certain aspects of our work.
> > >
> > > Regarding the novelty, we understand your reservations about combining several foundation models. However, we believe that the integration of these models in our context is not just a mere combination. In addition to utilizing foundational visual encoders and text encoders found in previous methods, our approach uniquely leverages image segmentation foundation models (such as SAM), image captioning foundation models (like BLIPV2 and Tag2Text), and CHATGPT. To the best of our knowledge, our method represents the first exploration of these foundation models in the realm of 3D scene understanding.
> > >
> > > Regarding the similarity to ULIP, it is important to highlight some key differences that set our method apart. While the training of ULIP relies on the annotation of 3D objects to extract corresponding text features, this approach is not applicable within 3D scene datasets. Our method uniquely overcomes this limitation by employing image-to-text foundation models to generate relevant text descriptions, further refined using CHATGPT. Moreover, we acknowledge that our method, like ULIP, utilizes both text and image during the pre-training stage. However, it's essential to recognize the distinctions between those two methods. ULIP's instance-level contrastive approach serves well for tasks like instance discrimination but falls short for dense prediction tasks such as object detection and semantic segmentation. This limitation has been observed in various works across 2D and 3D domains [1, 2, 3, 9, 53]. There exists a non-negligible gap between instance-level pre-training as ULIP and our proposed scene-level dense pre-training method.
> > >
> > > The novelty of our work lies in the development of self-supervised pre-training that is tailored and aligned with 3D object detection and segmentation. This alignment is achieved through the strategic integration of foundation models from both image and text modalities, forming a coherent framework that transcends previous limitations in the field.
> > >
> > > We sincerely appreciate your feedback and the increased score. Your insights have been instrumental in refining our work, and we're grateful for your constructive engagement with our research.
> > >
> > > [1] Xie, Zhenda, et al. "Propagate yourself: Exploring pixel-level consistency for unsupervised visual representation learning." Proceedings of the IEEE/CVF Conference on Computer Vision and Pattern Recognition. 2021.
> > >
> > > [2] Wang, Xinlong, et al. "Dense contrastive learning for self-supervised visual pre-training." Proceedings of the IEEE/CVF Conference on Computer Vision and Pattern Recognition. 2021.
> > >
> > > [3] Liu, Yueh-Cheng, et al. "Learning from 2d: Contrastive pixel-to-point knowledge transfer for 3d pretraining." arXiv preprint arXiv:2104.04687 (2021).Liu, Yueh-Cheng, et al. "Learning from 2d: Contrastive pixel-to-point knowledge transfer for 3d pretraining." arXiv preprint arXiv:2104.04687 (2021).

---

### Official Review · Reviewer_hzsQ · 2023-07-06

**Soundness:** 2 fair
**Presentation:** 3 good
**Contribution:** 2 fair
**Rating:** 6
**Confidence:** 4

**Summary:**

This paper aims to improve the pre-training of 3D point cloud models by leveraging various foundational models, including vision models such as SAM and Grounding Dino, language models like ChatGPT, and vision-language models. To achieve this objective, the paper proposes several techniques:
1. Generating instance segmentation masks for the corresponding 2D images of 3D point clouds. This is accomplished by first generating image captions using caption models and language models like Tag2text and ChatGPT. Then, vision models such as SAM and Grounding Dino are employed to generate masks based on these captions.
2. Masked point cloud reconstruction with the masks obtained in the previous step.
3. Scene-level distillation using the features of an image encoder (2D-3D Disti) and a language encoder (Text-3D Distill).
4. Object-level distillation using the features of an image encoder (2D-3D Disti) and a language encoder (Text-3D Distill).
The proposed pre-training method is evaluated on several downstream tasks, including ScanNetV2 and SUN-RGBD object detection, as well as S3DIS semantic segmentation.

**Strengths:**

1. The idea of utilizing foundational models for 3D point cloud modeling is both interesting and promising.
2. The paper is well-written and easy to follow.
3. The experimental results consistently demonstrate improvements in several downstream tasks when compared to the baseline.

**Weaknesses:**

1. My main concern is that the experimental results are not sufficiently convincing. Although the results demonstrate consistent improvements compared to the baselines, the performance of the baselines used in this paper appears to be relatively low (e.g., the results of 3DETR and GroupFree3D). Additionally, the final system's performance lags behind the current state-of-the-art methods.
2. The overall pipeline is complex, making it difficult to evaluate the effectiveness of each component, such as the model selection for each component.
3. typos: (1) On line 159, the reference 'MAE [63]' is incorrect. (2) On line 233, it should be 'text branch' instead of 'image branch' when mentioning the utilization of CLIP ViT-B [48] to extract text features. (3) The teaser of Figure 2 is somewhat confusing.

**Questions:**

See Weaknesses

**Limitations:**

See Weaknesses

---

> ### Author Rebuttal · Authors · 2023-08-09
>
> With gratitude for the reviewer's valuable insights, we will make appropriate revisions to our paper. In the following section, we address each of the reviewer's questions and comments in detail.
>
>
> **W1: Experimental results are not sufficiently convincing**
>
> **1, Compare with the best performance of GroupFree3D.** We explored our method's efficiency by adjusting the plain transformer to ViT-B configurations, allowing a direct comparison with the PointNet++x2 settings in GroupFree3D. This novel approach led us to surpass GroupFree3D's performance with PointNet++w2× settings, as evident in Table 1. Remarkably, the results on the SUN RGB-D dataset have reached **state-of-the-art** performance. Though we are optimistic that using ViT-L configurations could further elevate our performance in ScanNet, our resources' limitations have hindered such experimentation for now.
>
> **Table 1: 3D object detection results on ScanNet and SUN RGB-D dataset based on GroupFree3D.**
>
> | Methods              | Encoder        | Pre-trained | SUN RGB-D $AP_{25}$ | SUN RGB-D $AP_{50}$ | ScanNetV2 $AP_{25}$ | ScanNetV2 $AP_{50}$ |
> |----------------------|----------------|-------------|---------------------|---------------------|---------------------|---------------------|
> | GroupFree3D          | PointNet++w2×  | *None*      | -                   | -                   | 69.1                | 52.8                |
> | +plain transformer   | ViT-B          | *None*      | 65.2                | 47.1                | 68.6                | 51.9                |
> | +Bridge3D            | ViT-B          | ✓           | **69.6**            | **52.6**            | **71.2**            | **54.4**            |
>
> **2, Apply Bridge3D in SOTA methods.**  Our method's adaptability extends to its successful application to the recent state-of-the-art (SOTA) work, CAGroup3D [1]. While the network structure made a direct application of the plain transformer challenging, we devised a unique adaptation, pre-training the CAGroup3D's backbone with our scene and object-level distillation, excluding the reconstruction part. Table 2 illustrates our pre-training approach can enhance and benefit current SOTA 3D detection methodologies, showcasing the potential reach of our technique. Importantly, it should be noted that our framework is optimized for plain transformer-based backbones, and thus the application of Bridge3D to alternative backbones will lead to a reduction in performance.
>
> **Table 2: 3D object detection results on ScanNet and SUN RGB-D dataset based on CAGroup3D.**
>
> | Methods                                               | Pre-trained |SUN RGB-D $AP_{25}$ | SUN RGB-D $AP_{50}$ | ScanNetV2 $AP_{25}$ | ScanNetV2 $AP_{50}$ |
> |-------------------------------------------------------|-------------|-----------|-----------|-----------|-----------|
> | CAGroup3D [1]                                         | *None*      | 66.8      | 50.2      | 75.1      | 61.3      |
> | +Bridge3D (scene & object level distillation)         | ✓           | **68.7**  | **52.1**  | **76.8**  | **62.6**  |
>
> **W2: The complexity of the framework**.
>
> The complexity observed in our approach is primarily due to our full utilization of modern foundation models across both image and text modalities. This integration is vital to capture the intricate relationships inherent in multi-modal modeling and fully utilize foundation models for 3D representation learning. However, the complexity is not an immovable feature of our model. Our framework consists of three distinct components and two teacher modalities, all designed independently. The ablation study in Table 3 of the main paper reveals the positive contribution of each component to the final results, justifying the flexibility. While this robust design may initially seem daunting, it is strategically crafted to prevent complexity from becoming a barrier. For specific use cases, users have the flexibility to select and combine different components and modalities, enabling them to harness the full capabilities of our method or decrease complexity according to their requirements. As shown in Figure 2 in the main paper, it is convenient to decrease modalities or components by just removing the corresponding distillation loss. Our method is a powerful and flexible framework capable of achieving state-of-the-art results without undue complexity or difficulty in implementation.
>
> **W3: Typos**
>
> Thank you for meticulously identifying the specific typos and inconsistencies in our manuscript. Your attention to detail greatly aids in enhancing the accuracy and clarity of our work. Regarding the MAE [63] reference on line 159, we will correct it as I2P-MAE  in the revised version. On line 233, your observation is indeed accurate. It should be 'text branch' instead of 'image branch' when referring to the utilization of CLIP ViT-B [48] for text feature extraction. We will amend this error in the final manuscript. We recognize that the teaser of Figure 2 may be confusing, and we will carefully redesign it for better understanding.
>
>
> [1] Wang, Haiyang, et al. "Cagroup3d: Class-aware grouping for 3d object detection on point clouds." Advances in Neural Information Processing Systems 35 (2022): 29975-29988.

---

> > ### Author Response · Authors · 2023-08-15
> >
> > **Extra response to the concern on complexity.** As demonstrated in Table 1, the results indicate that each modality contributes to better results. When we combine all modalities, our proposed method achieves the best result. Furthermore, our method demonstrates robustness to variations in the choice of modalities. This flexibility in modal choice means that the complexity of the framework can be scaled down without significant degradation in performance. In essence, users can tailor the framework to their specific needs, allowing for a more adaptable application of our method without sacrificing quality.
> >
> > **Table 1: The effectiveness of each modality. Ablation study on the effectiveness of each modality on 3D object detection and semantic segmentation tasks**.
> > | Image Modality | Text Modality | ScanNetV2 $AP_{25}$ | ScanNetV2 $AP_{50}$ | S3DIS $mIoU$ | S3DIS $mAcc$ |
> > |----------------|---------------|---------------------|---------------------|--------------|--------------|
> > |                |               | 62.1                | 37.9                | 61.1         | 67.2         |
> > | ✔️            |               | 65.7                | 44.6                | 69.0         | 75.4         |
> > |                | ✔️           | 64.8                | 43.2                | 68.4         | 75.2         |
> > | ✔️            | ✔️           | **66.3**            | **45.5**            | **71.1**     | **77.5**     |

---

> > ### Comment · Reviewer_hzsQ · 2023-08-16
> > **Reply to the rebuttal**
> >
> > Most of my concerns are well addressed after reading the rebuttal. I have changed my Rating to Weak Accept.

---

> > > ### Author Response · Authors · 2023-08-17
> > > **Thanks for the revised score**
> > >
> > > We sincerely appreciate your careful review and consideration of our rebuttal. It's encouraging to hear that we have been able to address most of your concerns, leading to an updated rating.

---

### Official Review · Reviewer_E1Dc · 2023-07-06

**Soundness:** 2 fair
**Presentation:** 2 fair
**Contribution:** 3 good
**Rating:** 6
**Confidence:** 4

**Summary:**

a novel masking and patch-dropping strategy based on foundation models to refine the focus of the network on foreground representation learning for 3D masked autoencoders.

a novel scene-level and object-level multi-modality knowledge distillation method that pre-trains a 3D network via features, semantic masks, and captions obtained from foundation models

**Strengths:**

Personally I really like the self-supervised framework. It's natural to use MAE to reconstruct and build 3D representations, and obtaining 3D features from 2D images and text is also reasonable. The segmentation results seem good.


**Weaknesses:**

* Lack of qualitative examples. The biggest problem to me is that no qualitative examples are presented in the paper (and visualization is really important for 3D reconstruction and segmentation!)
* The experiments focus on segmentation. However, there's more to do with foundation models - like showing zero-shot results, captioning (generation or retrieval) etc.
* The paper doesn't mention several recent papers on 3D representation / feature / concept learning from 2D features (like CLIP). Just to name a few:

ConceptFusion: Open-set Multimodal 3D Mapping

Semantic Abstraction: Open-World 3D Scene Understanding from 2D Vision-Language Models

3D Concept Learning and Reasoning from Multi-View Images


This paper is also really really similar:

ULIP: Learning a Unified Representation of Language, Images, and Point Clouds for 3D Understanding

Also, I think the self-supervision and feature learning framework shares merits with this paper:
3D Concept Grounding on Neural Fields.

Some of these methods could serve as baselines for segmentation. If not, you *should at least cite a few papers in the domain*.



**Questions:**

Misc:
The *Mask*s mentioned in 3D representation and in 2D segmentation are two different things and may seem confusing. It's better if you could replace one of them by other words (like occupancy for 3D)

**Limitations:**

see weaknesses

---

> ### Author Rebuttal · Authors · 2023-08-09
>
> With gratitude for the reviewer's valuable insights, we will make appropriate revisions to our paper. In the following section, we address each of the reviewer's questions and comments in detail.
>
>
>
> **W1: Lack of visualization results**
>
> Thank you for the valuable feedback regarding the absence of visualization for segmentation and reconstruction in our paper. We put visualization results of reconstruction, segmentation, and detection in the attached PDF file in the global response.
>
>
> **W2: More to do with foundation models**
>
> Thank you for your insightful feedback. Currently, our focus is on leveraging foundation models to advance 3D detection and segmentation. While we recognize the potential in broadening our approach to encompass areas such as 3D zero-shot results and captioning, the constraints of the rebuttal period prevent us from diving into these tasks immediately. In the future, we will explore those avenues. Your perspective is invaluable and provides clear direction for expanding the horizons of our research. We're grateful for your thoughtful suggestions.
>
>
> **W3: Cite more papers**
>
> Thank you for bringing to our attention the absence of references to recent papers on 3D representation, feature, and concept learning from 2D features.
>
> **1, Citation.** We acknowledge that these works form important contexts and could provide additional insights into our work. Currently, our focus is primarily on leveraging foundation models for 3D detection and segmentation, and it appears we inadvertently omitted some relevant recent developments. We are grateful for your feedback, and we will add and discuss them in relative works.
>
>
> **2, Comparison.** It's worth noting that we have included a comparison with a recent CVPR 2023 paper, CLIP2Scene [9], in our ablation study. This paper proposes the use of MaskCLIP [69] to generate 2D masks and distill corresponding object-level text features into 3D, which is similar to the paper Semantic Abstraction [2]. By applying it to our framework and comparing it, we found that our method outperforms it in both 3D detection and segmentation tasks.
>
> **3, Difference to the ULIP [1].** The ULIP method is specifically tailored for pre-training on instance-level datasets such as ModelNet40 and ScanObjectNN, and it is hard to be directly applied to 3D scene understanding tasks. Furthermore, our method overcomes the limitations of ULIP that need text labels to generate text features. We leverage powerful image captioning models to generate scene text descriptions and utilize CHATGPT to refine these texts without requiring annotations.
>
>
> **W4: Replace 2D/3D masks by other words**
>
> Thank you for pointing out the potential confusion regarding the use of the term "Masks" in both 3D representation and 2D segmentation within our manuscript. Your observation highlights an important subtlety that we may have overlooked. We will differentiate these concepts by using distinct terms, 3D occupancy in the final version.
>
> [1] Xue, Le, et al. "ULIP: Learning a unified representation of language, images, and point clouds for 3D understanding." Proceedings of the IEEE/CVF Conference on Computer Vision and Pattern Recognition. 2023.
> [2] Ha, Huy, and Shuran Song. "Semantic abstraction: Open-world 3d scene understanding from 2d vision-language models." 6th Annual Conference on Robot Learning. 2022.

---

> > ### Comment · Reviewer_E1Dc · 2023-08-19
> >
> > I have read your response. Thank you for the segmentation results. I will keep my original score(6).

---

### Official Review · Reviewer_XLcq · 2023-07-06

**Soundness:** 3 good
**Presentation:** 3 good
**Contribution:** 3 good
**Rating:** 6
**Confidence:** 4

**Summary:**

This paper proposes a combination of several techniques to pre-train 3D models by leveraging 2D foundation models. Specifically, the paper proposes: a) an MAE pre-training guided by foreground masks generated by foundation models to mask the foreground masks more compared to the background, to encourage the models to focus more on the foreground. b) Scene-level feature distillation between 3D-image and 3D-text with the help of image captioning foundation model c) Object level feature distillation with the help of mask generated via foundation models. The results show better performance compared to prior methods of self-supervising the 3D models and the ablations confirm the benefit of each proposed technique.

**Strengths:**

The paper is overall well-written and the results are impressive compared to the baselines. The ablations are clear and well-motivated. Investigating how far we can get by leveraging 2D foundation model is an important research direction, and this paper does a good job in exploring several ways to go about it.

**Weaknesses:**

I do not have any substantial complaint with this paper, however, I think the main limitation is in the complexity of the proposed technique: the approach has several moving components relying on several off-the-shelf models and it seems like a combination of a bunch of techniques to utilize the 2D foundation models; requiring several offline computation of intermediate results.

**Questions:**

N/A, I do any have any specific questions.

**Limitations:**

No, I think it would be useful to specify limitations in the main paper (for eg: complexity of the approach)

---

> ### Author Rebuttal · Authors · 2023-08-09
>
> With gratitude for the reviewer's valuable insights, we will make appropriate revisions to our paper. In the following section, we address each of the reviewer's questions and comments in detail.
>
> **W1: The complexity.**
>
> **1, Complexity of the framework.** The complexity observed in our approach is primarily due to our full utilization of modern foundation models across both image and text modalities. This integration is vital to capture the intricate relationships inherent in multi-modal modeling and fully utilize foundation models for 3D representation learning. However, the complexity is not an immovable feature of our model. Our framework consists of three distinct components and two teacher modalities, all designed independently. The ablation study in Table 3 of the main paper reveals the positive contribution of each component to the final results, justifying the flexibility. While this robust design may initially seem daunting, it is strategically crafted to prevent complexity from becoming a barrier. For specific use cases, users have the flexibility to select and combine different components and modalities, enabling them to harness the full capabilities of our method or decrease complexity according to their requirements. As shown in Figure 2 in the main paper, it is convenient to decrease modalities or components by just removing the corresponding distillation loss. Our method is a powerful and flexible framework capable of achieving state-of-the-art results without undue complexity or difficulty in implementation.
>
> **2, Complexity of the computation.** It's worth noting that although our method incorporates several off-the-shelf foundation models to obtain masks, captions, and features, this complexity is somewhat mitigated by the nature of our approach. The computation involving these models is performed offline and only once. During both the training and inference stages, these foundation models are not used, meaning they do not substantially increase the complexity of computation. Such preprocessing is common in self-supervised learning methods like SLidR [53] and CLIP2Scene [9] mentioned in the main paper. Thus, while the initial setup may seem complex, it is in line with established methodologies and does not unduly burden the ongoing computational process. We believe that this carefully designed architecture enables us to harness the power of these foundation models without unnecessary complications in deployment.
>
> **W2: Limitations not discussed.**
>
> Thank you for your thoughtful suggestion. We acknowledge that a comprehensive discussion of the limitations of our approach, including its complexity, is paramount to a well-rounded understanding. In the following section, we will delve into these limitations and outline potential avenues for future work.
>
> While our work has achieved promising results in 3D detection and segmentation tasks, it encounters specific limitations that merit further exploration. A significant constraint is the method's reliance on existing foundation models for mask and caption generation, leading to several offline computations of intermediate results and time-consuming data preprocessing. This dependency also means that changing or updating these underlying models might affect overall performance. In the future, we aim to decrease the computation complexity by integrating high-speed models, such as mobile SAM [1], and create a more balanced and efficient system that can maintain high performance without being heavily tied to the pre-existing models currently in use.
>
> [1] Zhao, X., Ding, W., An, Y., Du, Y., Yu, T., Li, M., ... & Wang, J. (2023). Fast Segment Anything. arXiv preprint arXiv:2306.12156.

---

> > ### Comment · Reviewer_XLcq · 2023-08-10
> >
> > I have read your response, thank you for your detailed explanation!

---

### Official Review · Reviewer_V9nH · 2023-07-07

**Soundness:** 3 good
**Presentation:** 2 fair
**Contribution:** 3 good
**Rating:** 5
**Confidence:** 4

**Summary:**

This paper proposes a methodology named Bridge3D to pre-train 3D models using features, semantic masks, and captions sourced from foundation models. Experiments have been conducted on different 3D understanding frameworks.

**Strengths:**

The overall architecture seems to be reasonable; using the modern foundation models and multimodal supervision is promising to help 3D understanding, and some of the results show a good improvement gap compared to training from scratch.

**Weaknesses:**

This framework seems to be overall complicated, which might involve quite some tuning effort and impact its scalability.

**Questions:**

The reviewer is curious whether the authors have tried to simplify this framework.
Another question is that, for the 3D object detection results, table 1, it seems only the basic versions of the models have been tried. For example, the GroupFree3D, the best performance on ScanNetV2 reported in this paper is 69.7 and 52.4, however, in the GroupFree3D original paper, the best performance they reported is 69.1 and 52.8, which is basically on par with the reported results in this paper, the 3DETR results have the same issue. Have the authors tried to apply your method to their most powerful models? What is the highest performance you may get in that case, will that reach new SOTA results? So far the improvement compared to the vanilla baseline looks fine but in order to demonstrate the proposed method to be more useful, the absolute performance or the highest performance by applying the proposed method needs to be explored.

**Limitations:**

Refer to the weakness part.

---

> ### Author Rebuttal · Authors · 2023-08-08
>
> With gratitude for the reviewer's valuable insights, we will make appropriate revisions to our paper. In the following section, we address each of the reviewer's questions and comments in detail.
>
> **W1: The complexity of the framework.**
>
> **Concerns on complexity.**
> The complexity observed in our approach is primarily due to our full utilization of modern foundation models across both image and text modalities. This integration is vital to capture the intricate relationships inherent in multi-modal modeling and fully utilize foundation models for 3D representation learning. However, the complexity is not an immovable feature of our model. Our framework consists of three distinct components and two teacher modalities, all designed independently. The ablation study in Table 3 of the main paper reveals the positive contribution of each component to the final results, justifying the flexibility. While this robust design may initially seem daunting, it is strategically crafted to prevent complexity from becoming a barrier. For specific use cases, users have the flexibility to select and combine different components and modalities, enabling them to harness the full capabilities of our method or decrease complexity according to their requirements. As shown in Figure 2 in the main paper, it is convenient to decrease modalities or components by just removing the corresponding distillation loss. Our method is a powerful and flexible framework capable of achieving state-of-the-art results without undue complexity or difficulty in implementation.
>
> **Concerns on tuning efforts.** Our approach is not as tuning-intensive as it may appear. Many elements of mask and caption generation rely on pre-existing foundation models that are frozen and require no additional tuning. In terms of training, our method only introduces one additional parameter related to the foreground masking ratio. This extra parameter does not significantly impact scalability, particularly when compared to the baseline method, Point-MAE.
>
> **W2: Apply Bridge3D to more powerful models.**
>
> **1, Compare with the best performance of GroupFree3D.** We explored our method's efficiency by adjusting the plain transformer to ViT-B configurations, allowing a direct comparison with the PointNet++x2 settings in GroupFree3D. Our novel approach led us to surpass GroupFree3D's performance with PointNet++w2× settings, as shown in Table 1. Remarkably, the results on the SUN RGB-D dataset have reached **state-of-the-art** performance. Though we are optimistic that using ViT-L configurations could further elevate our performance in ScanNet, our resources' limitations have hindered such experimentation for now.
>
> **Table 1: 3D object detection results on ScanNet and SUN RGB-D dataset based on GroupFree3D.**
>
> | Methods              | Encoder        | Pre-trained | SUN RGB-D $AP_{25}$ | SUN RGB-D $AP_{50}$ | ScanNetV2 $AP_{25}$ | ScanNetV2 $AP_{50}$ |
> |----------------------|----------------|-------------|---------------------|---------------------|---------------------|---------------------|
> | GroupFree3D          | PointNet++w2×  | *None*      | -                   | -                   | 69.1                | 52.8                |
> | +plain transformer   | ViT-B          | *None*      | 65.2                | 47.1                | 68.6                | 51.9                |
> | +Bridge3D            | ViT-B          | ✓           | **69.6**            | **52.6**            | **71.2**            | **54.4**            |
>
> **2, Apply Bridge3D in SOTA methods.**  Our method's adaptability extends to its successful application to the recent state-of-the-art (SOTA) work, CAGroup3D [1]. While the network structure made a direct application of the plain transformer challenging, we devised a unique adaptation, pre-training the CAGroup3D's backbone with our scene and object-level distillation, excluding the reconstruction part. Table 2 illustrates our pre-training approach can enhance and benefit current SOTA 3D detection methodologies, showcasing the potential reach of our technique. Importantly, it should be noted that our framework is optimized for plain transformer-based backbones, and thus the application of Bridge3D to alternative backbones may lead to a reduction in performance.
>
> **Table 2: 3D object detection results on ScanNet and SUN RGB-D dataset based on CAGroup3D.**
>
> | Methods                                               | Pre-trained | SUN RGB-D $AP_{25}$ | SUN RGB-D $AP_{50}$ | ScanNetV2 $AP_{25}$ | ScanNetV2 $AP_{50}$ |
> |-------------------------------------------------------|-------------|-----------|-----------|-----------|-----------|
> | CAGroup3D [1]                                         | *None*      | 66.8      | 50.2      | 75.1      | 61.3      |
> | +Bridge3D (scene & object level distillation)         | ✓           | **68.7**  | **52.1**  | **76.8**  | **62.6**  |
>
> [1] Wang, Haiyang, et al. "Cagroup3d: Class-aware grouping for 3d object detection on point clouds." Advances in Neural Information Processing Systems 35 (2022): 29975-29988.

---

> > ### Author Response · Authors · 2023-08-15
> >
> > **Extra response to the concern on complexity.** As demonstrated in Table 1, the results indicate that each modality contributes to better results. When we combine all modalities, our proposed method achieves the best result. Furthermore, our method demonstrates robustness to variations in the choice of modalities. This flexibility in modal choice means that the complexity of the framework can be scaled down without significant degradation in performance. In essence, users can tailor the framework to their specific needs, allowing for a more adaptable application of our method without sacrificing quality.
> >
> > **Table 1: The effectiveness of each modality. Ablation study on the effectiveness of each modality on 3D object detection and semantic segmentation tasks**.
> > | Image Modality | Text Modality | ScanNetV2 $AP_{25}$ | ScanNetV2 $AP_{50}$ | S3DIS $mIoU$ | S3DIS $mAcc$ |
> > |----------------|---------------|---------------------|---------------------|--------------|--------------|
> > |                |               | 62.1                | 37.9                | 61.1         | 67.2         |
> > | ✔️            |               | 65.7                | 44.6                | 69.0         | 75.4         |
> > |                | ✔️           | 64.8                | 43.2                | 68.4         | 75.2         |
> > | ✔️            | ✔️           | **66.3**            | **45.5**            | **71.1**     | **77.5**     |

---

> > ### Comment · Reviewer_V9nH · 2023-08-19
> > **Thanks for the rebuttal**
> >
> > The reviewer has carefully read the authors' rebuttal and appreciates the experiments, and would like to keep the original rating.

---

### Author Rebuttal · Authors · 2023-08-09

We appreciate the time and effort spent reviewing our submission. In this response, we address the major concerns raised by the reviewers and respond to other concerns in individual rebuttal responses.

**Reviewer XLcq, MtnQ: The limitations of the paper not discussed**

Thank you for your thoughtful suggestion. The following part is about limitations and future work.

While our work has achieved promising results in 3D detection and segmentation tasks, it encounters specific limitations that merit further exploration. A significant constraint is the method's reliance on existing foundation models for mask and caption generation, leading to several offline computations of intermediate results and time-consuming data preprocessing. This dependency also means that changing or updating these underlying models might affect overall performance. In the future, we aim to investigate the integration of high-speed models, such as mobile SAM, and create a more balanced and efficient system that can maintain high performance without being heavily tied to the pre-existing models currently in use.

**Reviewer MtnQ: The Limitation of evaluation**

Thank you for your detailed insights into the evaluation section of our paper. We admit that the plain transformer-based compared method is not enough in the main paper. In Table 1, we add more transformer-based pre-training methods in 3D object detection tasks. Currently, due to the difference in backbones, transformer-based pretraining methods in 3D classification, detection, and segmentation (e.g. ACT [4], PiMAE [8], and MaskPoint [3]) only compare to other transformer-based methods. Our updated analysis confirms that Bridge3D outperforms existing transformer-based 3D pretraining methods, reinforcing our main conclusions.

**Table 1: 3D object detection results on ScanNet and SUN RGB-D dataset based on 3DETR.**

| Methods                 | Pre-trained | SUN RGB-D $AP_{25}$ | SUN RGB-D $AP_{50}$ | ScanNetV2 $AP_{25}$ | ScanNetV2 $AP_{50}$ |
|-------------------------|-------------|---------------------|---------------------|---------------------|---------------------|
| 3DETR                   | *None*      | 58.0                | 30.3                | 62.1                | 37.9                |
| +Bridge3D(from scratch) | *None*      | 57.6                | 31.9                | 61.1                | 38.6                |
| +Point-Bert[1]          | ✓           | -                   | -                   | 61.0                | 38.3                |
| +Point-MAE [44]         | ✓           | 59.2                | 33.2                | 62.3                | 39.9                |
| +Point-M2AE [2]         | ✓           | -                   | -                   | 63.7                | 40.8                |
| +MaskPoint [3]          | ✓           | -                   | -                   | 63.4                | 40.6                |
| +PiMAE [8]              | ✓           | 59.9                | 33.7                | 63.0                | 40.2                |
| +ACT [4]                | ✓           | -                   | -                   | 63.5                | 41.0                |
| +Bridge3D               | ✓           | **62.5(+4.9)**      | **36.8(+4.9)**      | **66.3(+5.4)**      | **45.5(+6.2)**      |



However, we agree that our paper could benefit from a more comprehensive comparison. We applied our method to the VoteNet framework by replacing the PointNet++ architecture with the plain transformer architecture. The result in Table 2 from this adaptation demonstrates that Bridge3D exhibits superior performance.

**Table 2: 3D object detection results on ScanNet and SUN RGB-D dataset based on VoteNet.**

| Methods                 | Pre-trained | SUN RGB-D $AP_{25}$ | SUN RGB-D $AP_{50}$ | ScanNetV2 $AP_{25}$ | ScanNetV2 $AP_{50}$ |
|-------------------------|-------------|---------------------|---------------------|---------------------|---------------------|
| VoteNet                 | *None*      | 57.7                | 32.9                | 58.6                | 33.5                |
| +Bridge3D(from scratch) | *None*      | 59.6                | 34.9                | 60.8                | 37.3                |
| +PointContrast [28]     | ✓           | 57.5                | 34.5                | 59.2                | 38.0                |
| +Hou et al. [27]        | ✓           | -                   | 36.4                | -                   | 39.3                |
| +4DContrast [10]        | ✓           | -                   | 38.2                | -                   | 40.0                |
| +DepthContrast [65]     | ✓           | 61.6                | 35.5                | 64.0                | 42.9                |
| +DPCo [36]              | ✓           | 60.2                | 35.5                | 64.2                | 41.5                |
| +Bridge3D               | ✓           | **63.7(+4.1)**      | **39.5(+4.6)**      | **67.1(+6.3)**      | **44.3(+7.0)**      |







[1] Yu, X., Tang, L., Rao, Y., Huang, T., Zhou, J., & Lu, J. (2022). Point-bert: Pre-training 3d point cloud transformers with masked point modeling. In Proceedings of the IEEE/CVF Conference on Computer Vision and Pattern Recognition

[2] Zhang, Renrui, Ziyu Guo, Peng Gao, Rongyao Fang, Bin Zhao, Dong Wang, Yu Qiao, and Hongsheng Li. "Point-m2ae: multi-scale masked autoencoders for hierarchical point cloud pre-training." Advances in neural information processing systems 35 (2022)

[3] Liu, H., Cai, M., & Lee, Y. J. (2022, October). Masked discrimination for self-supervised learning on point clouds. In European Conference on Computer Vision. Cham: Springer Nature Switzerland.

[4] Dong, Runpei, et al. "Autoencoders as Cross-Modal Teachers: Can Pretrained 2D Image Transformers Help 3D Representation Learning?." The Eleventh International Conference on Learning Representations. 2022.

---

### Decision · Program_Chairs · 2023-09-21

**Decision:**

Accept (poster)

**Comment:**

In this paper, the authors propose Bridge3D, a method for 3D scene representation learning by leveraging recent foundational models, such as segmentation and captioning models. After the rebuttal, all reviewers found their concerns addressed and recommend acceptance. After reading all reviews and rebuttals, the AC agrees with the decision.